# The Impact of Focused Ultrasound in Two Tumor Models: Temporal Alterations in the Natural History on Tumor Microenvironment and Immune Cell Response

**DOI:** 10.3390/cancers12020350

**Published:** 2020-02-04

**Authors:** Gadi Cohen, Parwathy Chandran, Rebecca M. Lorsung, Lauren E. Tomlinson, Maggie Sundby, Scott R. Burks, Joseph A. Frank

**Affiliations:** 1Frank Laboratory, Radiology and Imaging Sciences, Clinical Center, National Institutes of Health, 10 Center Drive, Bethesda, MD 20892-1074, USA; parwathy.chandran@nih.gov (P.C.); Rebecca.Lorsung@som.umaryland.edu (R.M.L.); lauren.tomlinson@nih.gov (L.E.T.); maggie.sundby@nih.gov (M.S.); scott.burks@nih.gov (S.R.B.); 2National Institutes of Biomedical Imaging and Bioengineering, National Institutes of Health, 10 Center Drive, Bethesda, MD 20892-1074, USA

**Keywords:** B16 melanoma, 4T1 breast cancer, pulsed focused ultrasound, proteomic, tumor microenvironment, immune cells

## Abstract

Image-guided focused ultrasound (FUS) has been successfully employed as an ablative treatment for solid malignancies by exposing immune cells to tumor debris/antigens, consequently inducing an immune response within the tumor microenvironment (TME). To date, immunomodulation effects of non-ablative pulsed-FUS (pFUS) on the TME are poorly understood. In this study, the temporal differences of cytokines, chemokines, and trophic factors (CCTFs) and immune cell populations induced by pFUS were interrogated in murine B16 melanoma or 4T1 breast cancer cells subcutaneously inoculated into C57BL/6 or BALB/c mice. Natural history growth characteristics during the course of 11 days showed a progressive increase in size for both tumors, and proteomic analysis revealed a shift toward an immunosuppressive TME. With respect to tumor natural growth, pFUS applied to tumors on days 1, 5, or 9 demonstrated a decrease in the growth rate 24 h post-sonication. Flow cytometry analysis of tumors, LNs, and Sp, as well as CCTF profiles, relative DNA damage, and adaptive T-cell localization within tumors, demonstrated dynamic innate and adaptive immune-modulation following pFUS in early time points of B16 tumors and in advanced 4T1 tumors. These results provide insight into the temporal dynamics in the treatment-associated TME, which could be used to evaluate an immunomodulatory approach in different tumor types.

## 1. Introduction

Tumors exist as a complex heterogenous milieu comprising populations of neoplastic cells, stromal cells, immune cells, secreted factors, vasculature, and extracellular matrices (ECMs), collectively acknowledged as the tumor microenvironment (TME) [1,2,3,4]. The TME is influenced by an intricate set of interactions between the transformed cells and the immune system, orchestrated by expressions of damage-associated molecular patterns (DAMPs), cytokines, chemokines, trophic factors (CCTFs), cell adhesion molecules (CAMs), and cellular debris [5,6,7,8]. The balance between CCTFs and CAMs, as well as immune cells, will influence tumor progression by either favoring an anti-tumor immune response or immune tolerance and suppression in the TME [9]. Cytokines that exemplify an immunosuppressive TME include transforming growth factor-beta (TGFβ), interleukins (IL) 4, 10, and 13, and vascular endothelial growth factor (VEGF) [10,11]. Elevated expressions of these intra-tumoral immunosuppressors frequently occur along with an impaired expression of pro-inflammatory mediators such as tumor necrosis factor alpha (TNFα), interferon gamma-induced protein-10 (IP-10), interferon gamma (IFNγ), or inflammatory cytokines such as IL1α, IL1β, IL2, and IL17 [12]. The TME CCTF influences the composition of the immune cell populations that drive the balance either toward an anti-tumor, cytotoxic immune response including cytotoxic T cells (T_cyt_), T helper cells (T_h_), dendritic cells (DCs), and natural killer (NK) cells [13,14], or an immunosuppressive microenvironment dominated by regulatory T cells (T_reg_), myeloid-derived suppressor cells (MDSCs), or anti-inflammatory tumor-associated macrophages (TAMs) [15,16]. Various cellular and biological immunotherapies have revolutionized the treatment of human malignancies by shifting the balance from an immunosuppressive to an anti-tumor TME by modulating immune cell populations within solid tumors. However, these approaches are associated with limited therapeutic efficacy [17,18,19], and interventional approaches are still considered as a stand-alone treatment or in combination with chemotherapy and/or radiation therapy for many patients with solid tumors [20]. There is widespread interest in developing less invasive approaches that generate an immune response targeting tumor cells and ultimately achieving improved clinical outcomes [21]. Less invasive tumor ablation approaches (e.g., radiofrequency, cryoablation, high-intensity ultrasound) are used in treating both primary and metastatic disease [22,23,24] to decrease tumor burden. Ablation can release various antigens and DAMPs in the local TME. Immune cells exposed to the tumor debris and DAMPs could stimulate changes in the TME [25,26], facilitating an anti-tumor immune setting [27,28,29]. High-intensity focused ultrasound (HIFU), a non-invasive technique that deposits focused energy, is being actively investigated as an ablative treatment for benign and malignant tumors [30,31,32,33,34,35]. Image-guided pulsed focused ultrasound (pFUS) is a non-ablative technique employing mechanical radiation, and acoustic cavitation forces have been applied in preclinical and clinical studies to increase the expression of CCTFs and CAMs in targeted regions for up to 24–48 h [36,37,38,39,40]. However, there have been limited studies regarding the mechanisms involving anti-tumor immunomodulation following pFUS to the TME that also stimulates a systemic immune response [41]. Moreover, optimal pFUS parameters capable of eliciting an immunotherapeutic response within the TME have not been thoroughly investigated in different tumor types, or temporally over the natural history course of disease progression. Changes in the expression of CCTFs, CAMs, and immune cell population could serve as useful indicators in assessing clinical outcomes. We recently characterized changes in CCTFs, CAMs, and immune cell infiltration 24 h post-pFUS in murine B16 melanoma and 4T1 breast cancer models by increasing peak negative pressures (PNPs) when flank tumors were ~8 mm in size [41]. The TME response to sonication was heterogenous with paradoxical factors that either promoted or downregulated immune responses between tumor types when sonicated at a specific size. To date, explorations have not been conducted into how the molecular and cellular composition in the TME of differently sized tumors will undergo changes in response to therapeutic interventions such as pFUS. In this study, we evaluated the natural history of the CCTF and CAM changes in 4T1 and B16 flank tumors over a period of 11 days to determine how the TME was altered with tumor growth. We also investigated whether pFUS exposure at different times (days 1, 5, and 9) during tumor growth would result in molecular, immune cell, and histological changes within the TME. This study provides a macroscopic overview on the effects of non-ablative sonication on the TME and underlines the potential impact of interpreting results considering the timing of therapeutic interventions in different types of tumor models.

## 2. Results

This study examined the proteomic changes in B16 and 4T1 flank tumors with respect to their growth in size over time, demonstrating temporal shifts that could favor either an anti-tumor or an immunosuppressive TME. Following non-ablative pFUS, variability in molecular response of the TME was observed between tumor types and tumor growth at different time points. Flow cytometry (FACS) of the tumor revealed variable amounts of modulation in immune cell populations within the tumor, spleen (Sp), and regional lymph nodes (LNs) 24 h post-pFUS. Histological analysis between sonicated and control tumors did not show evidence of increased areas of hemorrhage or necrosis, while DNA damage and enumeration of CD4^+^ and CD8^+^ localization within B16 and 4T1 tumors exhibited alterations with time.

### 2.1. Natural History Progress of B16 or 4T1 Tumors TME

#### 2.1.1. Proteomic Profiling During the Natural History Growth of Naïve Tumors

The natural history growth pattern of murine B16 melanoma and 4T1 breast cancer flank tumors altered the TME, which was apparent on proteomic analysis of the tumors (Figure 1A,B). Tumors were harvested on days 1, 3, 5, 7, 9, and 11 starting with ~5 mm diameter masses considered as day 1. An overall increase in tumor volume was noticed for both types of tumors (Figure 1C,D). Changes in pro-inflammatory (anti-tumor) and anti-inflammatory (immunosuppressive) factors evaluated over time were normalized to day 1 tumors and presented as a heat map (Figure 1A,B). Raw data of CCTF and CAM values with B16 and 4T1 tumor growth are presented in Appendix A. B16 tumors (Figure 1A) showed a significantly decreased expression (*p* < 0.05 one-way analysis of variance (ANOVA)) of IL1α, IL1β, IL6, IL17, macrophage colony stimulating factor (M-CSF), granulocyte CSF (G-CSF), keratinocyte chemoattractant (KC), Eotaxin, tumor necrosis factor α (TNFα), Lipopolysaccharide binding protein induced CXC chemokine (LIX/CXCL5), monocyte chemoattractant protein 1 (MCP-1), monocyte inflammatory protein 2 (MIP2), intercellular adhesion molecule (ICAM), and vascular cell adhesion molecule (VCAM) at various time points over 11 days compared to day 1 tumors. Significant increases (*p* < 0.05 ANOVA) were detected at various time points in IL2, IL4, IL9, IL12p40, IP-10, IFNγ, monokine induced by gamma interferon (MIG), regulated on activation, normal T cell expressed and secreted (RANTES), tumor growth factor β (TGFβ), and vascular endothelial growth factor (VEGF). Over 11 days, increased or decreased fold changes were observed in RANTES, IFNγ, VCAM, MIG, and MIP1a compared to day 1, primarily occurring at day 7. In addition, 4T1 tumors (Figure 1B) exhibited a proteomic profile different from B16 tumors. There were significant decreases (*p* < 0.05 ANOVA) in expressions of IL2, IL6, IL9, IL10, IL12p40, and IFNγ, and increased expressions for G-CSF, VCAM, and TGFβ over all days compared to tumors on day 1. Other CCTFs, which showed significant variations over one or several days during the 11 day time-course, include IL1a, IL1b, IP10, M-CSF, KC, leukemic inducible factor (LIF), Eotaxin, MIG, ICAM, VCAM, and VEGF. There were undetectable levels of IL15, IL17, and GM-CSF in 4T1 tumors. IL12p70 expression was undetectable in both tumors. No significant differences were detected for IL10 and LIF in B16 tumors, and LIX, MIP-1b, and TNFα in 4T1 tumors. These results demonstrate the molecular heterogeneity of the TME between tumor types that are molded with tumor expansion.

#### 2.1.2. Flow Cytometry of Naïve Tumors

In order to determine how tumor size and the corresponding TME would exert changes in the immune response to pFUS, flow cytometry analysis (FACS) was performed on tumor samples, Sp, and regional LNs harvested on days 2, 6, and 10 in naïve controls (Appendix A). For the untreated B16 naïve tumors, immune cell populations inconsistently varied from day 2 through day 10 (Appendix A). T_h_ and T_reg_ cells peaked on day 6, whereas T_cyt_, M1, M2, and B cells did not show any specific patterns. DCs peaked in the tumor on day 10 and there was a time-dependent increase in MDSCs from days 2 to 10. Checkpoint inhibitory receptor cytotoxic T-lymphocyte-associated protein 4 (CTLA4) and programmed death ligand 1 (PDL1) continued to decrease over 10 days, while the percentage of programmed cell death protein 1 (PD1) cells remained essentially unchanged over 10 days. T_h_, T_reg_, T_cyt_, B, DCs, and MDSC populations in Sp and LNs of naïve animals on days 6 and 10 were significantly decreased (*p* < 0.05 ANOVA) compared to naïve animals on day 2 (Appendix A). M1 macrophages in Sp on day 6 (*p* < 0.05 ANOVA) and day 10, as well as M2 macrophages on day 6, showed an elevated expression. For the untreated 4T1 naïve tumors, there was an inconsistency in the immune cell population changes within the tumor from day 2 through day 10 (Appendix A). In comparison to the B16 naïve tumors, T_h_, T_reg_, T_cyt_, and M1 macrophages peaked on day 10 in the 4T1 tumors. Pro-tumorigenic markers reached maximum percentages within the tumor at different time points: M2 macrophages on day 2, MDSCs, CTLA4, and PD1 on day 6, and PDL1 on day 10. Sp and LNs from the naïve 4T1 mice also demonstrated large temporal variations in percentages of detected immune cell populations (Appendix A). Significantly higher numbers of T_h_ and DCs were observed in the Sp on day 2 (*p* < 0.05 ANOVA) compared to other time points. MDSC expression continued to increase with time over the 10 days. T-cell expression seemed to taper by day 10. Splenic M1, M2, and B cell numbers remained largely unchanged over the 10 days in the naïve cohort. Within the LNs, all immune cell populations except M1 macrophages and B cells showed a significantly increased percentage on day 2 (*p* < 0.05 ANOVA) compared to other time points. B cell expression peaked on day 6, while the M1 macrophages number was significantly decreased on that day. There were an insufficient number of NK cells within the 4T1 tumors, Sp, and LNs to determine changes in either naïve or sonicated tumors. Overall, the natural history of percent changes in immune cell populations in mice within untreated tumors and peripheral tissues underscores the multi-dimensional complications in comparing results from the differently sized treated tumors (Appendix A).

### 2.2. pFUS-Immunomodulatory Effects in Relation to Different Growth Stages of B16 and 4T1 Tumors

#### 2.2.1. Proteomic Response to pFUS on Different Days with Changes in TME of B16 and 4T1 Tumors

To determine whether the TME would change depending on tumor size and treatment time (i.e., day 1, 5, and 9), B16 (Figure 2A) and 4T1 (Figure 2B) tumors (*n* = 6/time point) were sonicated with 6 MPa at ~5 mm diameter (day 1) and on days 5 and 9, and proteomic analyses were performed 24 h post-pFUS on days 2, 6, and 10. Tumor volumes were significantly reduced (*p* < 0.05, ANOVA) on days 6 and 10 post-pFUS for B16 (Figure 2C) and 4T1 (Figure 2D). In addition, 4T1 tumors were significantly smaller in volume on day 2 post-sonication (Figure 2D) compared to the untreated time-matched controls. Twenty-four hours post-pFUS of B16 and 4T1 tumors, there was no consistent proteomic pattern of changes across time points within the TME (Figure 2). As observed in the natural history study in Figure 1, the variability of the proteomic responses with time was independent of the B16 and 4T1 tumor sizes and, therefore, pFUS treatment to differently sized tumors resulted in inconsistent changes in the CCTFs and CAMs in the TME. The lack of consistency in the molecular responses to pFUS treatment of the B16 or 4T1 tumors was apparent at all time points when compared to naïve control tumors. Examination of CCTFs and CAMs in naïve controls at days 2, 6, and 10 revealed similar appearances of the heat map when compared to the natural history study, which would support an overlap in the changes in TME with tumor size of both B16 (Appendix A) and 4T1 (Appendix A) tumors. The molecular response to pFUS at any of the time points (i.e., days 2, 6, or 10) in the B16 (Appendix A) or 4T1 (Appendix A) tumors demonstrated variation in the TME that may not be as much a function of the treatment effects versus the heterogeneity of the naïve matched control tumors used to normalized the data. Overall, the molecular responses to pFUS are highly variable and, therefore, it may not be possible to compare observations in the TME when treatment interventions are performed in differently sized tumors.

#### 2.2.2. Flow Cytometry Analysis of Sonicated Tumors on Different Days Following pFUS

With the aim of investigating the immunomodulatory effects of multiple pFUS treatments (on days 1, 5, or 9) in the B16 and 4T1 tumors (*n* = 6 mice/time point), Sp, LNs, and tumors were harvested 24 h post-FUS on days 2, 6, and 10. Heatmaps of immune cell populations and checkpoint markers are presented as fold changes for each cell type at days 2, 6, and 10 in relation to time-matched untreated controls (Figure 3). There were significant differences (*p* < 0.05 ANOVA) between the expression levels of examined cell populations compared to the naïve controls, as well as inconsistencies as to which tissue type exhibits changes at the three time points. pFUS-treated B16, Sp, and LNs (Figure 3A) on day 2 showed no remarkable change in percentages of cell types compared to untreated controls. Interestingly, on day 6, Sp from the sonicated mice showed prominent changes with significantly high numbers of T_h_, T_reg_, T_cyt_, M1, B cells, NK cells, DCs, and MDSCs (*p* < 0.05 ANOVA), which started to decrease by day 10, with few changes in the LNs. Tissues harvested on day 10 revealed significant increases in T_h_, T_reg_, T_cyt_, M1, M2, and DCs (*p* < 0.05 ANOVA) in the LNs along with increases in T_h_, T_reg_, DCs, and MDSCs in the Sp. For B16 tumors at day 2, except for high T_cyt_, there was essentially no change in percentages of immune cell types compared to the controls. B cells peaked in day 6 tumors, while M2 expression was significantly reduced (*p* < 0.05 ANOVA). Day 10 tumors exhibited significant increases in tumor-infiltrating T_h_ and T_cyt_ (*p* < 0.05 ANOVA) compared to naïve tumors. The checkpoint receptors (i.e., CTLA4, PD1) and PDL1 expression were either significantly (*p* < 0.05 ANOVA) increased (days 2 and 6) or decreased compared to naïve tumors. In contrast to the B16 tumors, the heatmap for the 4T1 sonicated tumors (Figure 3B) presented a qualitatively different pattern with inconsistent temporal changes in the immune cell populations. On day 2, significant decreases in splenic T_h_, T_reg_, T_cyt_, M1, M2, DCs, and MDSCs (*p* < 0.05 ANOVA) were observed compared to the naïve control. The LN samples on day 2 demonstrated little change except for the decrease in T_reg_. On day 6, changes to immune cell profile included a significant increase in T_h_, T_reg_, and DCs in the Sp along with increases in T_cyt_, M2, and DCs in the LNs. The immune cell influx seemed to taper in both Sp and LN by day 10. In addition, 4T1 tumors on day 2 demonstrated significant percent increases in T_h_, T_reg_, T_cyt_, and B cells with the decrease in DCs and MDSCs. T_cyt_ and B cells were significantly elevated with decreased T_reg_, M1 macrophages, CTLA4, PD1, and PDL1 in day 6 tumors. By day 10, 4T1 tumors showed a high influx of cells with significant fold changes in T_cyt_, M1, M2, B, DCs, and MDSCs along with CTLA4, PD1, and PDL1. The immune cell profile of 4T1 tumors was suggestive of a marked shift toward an anti-tumor TME by day 10, which may provide explanations to the decrease in pFUS-treated tumor volume (Figure 3B). The temporal shift in immune cell phenotypes within tumors and peripheral tissues following pFUS does not follow a customary pattern. Nevertheless, the changes may support a re-balancing toward an anti-tumor TME.

#### 2.2.3. Histological Evaluation of Tumor Infiltrating CD4^+^ T_h_ and CD8^+^ T-cells Localization and Relative DNA Damage in pFUS-Treated Tumors 

Ultrasound-guided pFUS at 6 MPa was targeted over the entirety of B16 and 4T1 flank tumors, and mice (*n* = 3 sections/tumor type/time point) were perfused and tissue was paraffin-embedded for histological analysis. Qualitative analysis of hematoxylin and eosin staining (H and E) of either naïve controls or pFUS-treated B16 and 4T1 tumors revealed no differences in the amount of hemorrhage or necrosis on days 2, 6, and 10 (Figure 4A–C). Sonication of the tumor types did not change their consistencies with the B16 tumors being pliable and soft, while the 4T1 tumor was firmer and usually encapsulated. H and E staining also demonstrated heterogenous cytoarchitecture when comparing sonicated tumors to controls. TUNEL-positive cells were primarily localized to the center of the tumors and surrounding areas of necrosis. Quantitative analysis of TUNEL 24 h post-sonication revealed a significant increase (*p* < 0.05 ANOVA, with multiple comparison) in the relative positive nuclei compared to the control in B16 tumors treated with pFUS on day 1 (Figure 4D) and in 4T1 tumors when treated on day 9 (Figure 4E).

Immunohistochemistry was also performed on these tumor sections to evaluate the presence of tumor-infiltrating CD4^+^ T_h_ and CD8^+^ T_cyt_ in the pFUS-treated tumors, compared to untreated control tumors (Figure 5). Localization of tumor-infiltrating CD4^+^ T-cells demonstrated inconsistent appearance in both B16 and 4T1 tumors following sonication. Significantly high levels of CD4^+^ cells were detected near the periphery and center of post-sonicated B16 tumors (Figure 5D) on days 2 and 10, while CD4^+^ cells were localized more at the center of 4T1 tumors on days 6 and 10 post-sonication, as well as the marginal area of pFUS-treated 4T1 tumors on day 10 (Figure 5F). Excluding B16 tumors sonicated on day 9, elevations in CD8^+^ cells were noticed at both the center and periphery regions of sonicated tumors compared to untreated time-matched controls (Figure 5E). Comparable levels of CD8^+^ localization in B16 tumors on day 10 indicate the presence of an intrinsic inflammatory response within the TME, despite pFUS. In contrast to B16 tumors, the CD8^+^ cells localization was significantly higher in the center and margin areas of 4T1 tumors on days 6 and 9 post-sonication (Figure 5G). The amount of DNA damage between the sonicated and control tumor with the relative localizations of CD8^+^ cells suggest that pFUS at 6 MPa results in a significant change toward an anti-tumoral TME (*p* < 0.05 ANOVA) in pFUS-treated B16 tumors at early time points and in more advanced pFUS-treated 4T1 tumors. 

## 3. Discussion

The major findings in this study are that the molecular changes within the TME vary with tumor growth and tumor type. Furthermore, non-ablative pFUS at 6 MPa in differently sized tumors resulted in significantly slower tumor growth. This could be associated with varying molecular and cellular responses within the TME along with changes in cellular profiles in the Sp and regional LNs. The molecular and cellular responses to pFUS in differently sized tumors may possibly have been anticipated and underscores the potential difficulties in comparing therapeutic outcomes by attributing changes to the sonication without considering the underlying shift in responsiveness within the baseline TME (i.e., natural history of the disease). The TME has been described on a spectrum ranging from an immunosuppressive (cold) or an anti-tumor (hot) setting depending on the presence or concentration of pro-inflammatory or anti-inflammatory CCTFs and CAMs, as well as immune cell phenotypes, within a dynamically changing temporal landscape [8,42]. Cold tumors tend to express immunosuppressive CCTFs such as of TGFβ, IL-10, IL-4, IL-13, VEGF, RANTES, MCP-1, MCSF accompanied by increased infiltration of T_reg_, M2 macrophages, and MDSCs, with few functional DCs or antigen-presenting cells (APCs) [43]. The tumor vasculature expression of CAMs is also limited, thereby reducing the infiltration of peripheral immune cell populations into the TME [44]. Hot tumors (anti-tumor TME) tend to have increased expression of IFNγ, IL-1α, IL-1β, IL-2, IL-6, IL-12, IL-17, TNFα, IP-10, MIG, and CAMs [45,46] along with activation of immune cell populations including T_cyt_, NK cells, T_h_, T_h_17 cells, DCs, and functional APC, to generate an anti-tumor immune response and slow tumor growth [47]. Several CCTFs can demonstrate stimulatory or inhibitory effects on tumor development and progression depending on immune-suppressive or anti-tumor molecular profiles within the TME. Moreover, tumors originating from various organs are likely to contain contributions of cold and hot phenotypic zones, which might be influenced by external physical stimuli in divergent or similar patterns [48,49,50].

In the current study, we delineated the temporal molecular changes associated with tumor growth, which, albeit limited in scope, contributes to the important understanding of the natural history changes in the TME in untreated B16 and 4T1 tumors. There is a relative lack of investigation dedicated to characterizing the alterations associated with the dynamic and heterogeneous TME proteomic profiles when comparing small day 1 tumors (~5 mm diameter) to later-stage day 11 tumors with a spectrum from low to high fold changes in the heatmaps over time. By normalizing the CCTF and CAM expression to different days during tumor expansion, it becomes apparent that at any given time point, there are complex processes at work influencing the status (i.e., hot or cold) of the TME [51]. The communication of the tumor cells with the TME includes the autocrine/paracrine release of CCTFs, proteases, bioactive molecules, extracellular matrix (ECM) remodeling, cancer-associated fibroblast (CAF), immune cell populations, exosomes, areas of hypoxia, acidosis, nutritional demands, genetic drift, and enhanced anti-apoptotic signaling along with vasculogenesis, which will allow for the tumor expansion in the absence of treatment [51,52,53,54]. The natural history changes in CCTFs and CAMs in both B16 and 4T1 tumors revealed that with growth progression, there was a shift toward an immunosuppressive or cold TME. Early in the proliferative growth phase of B16 and 4T1 tumors, there were increases in local pro-inflammatory CCTF and CAM expressions when compared to larger tumors. However, further studies are still needed in identifying CCTF and CAM changes derived from specific cell populations in the TME at different time points associated with tumor growth.

We previously reported proteomic and immune cell changes in the TME resulting in non-specific molecular profiles following pFUS at 24 h at various PNPs [41]. pFUS performed at 6 MPa resulted in increased DNA damage and associated changes in CCTFs that could result in immunomodulation of the TME [41]. Of particular interest is DNA damage observed by TUNEL assays. DNA damage alone generates proinflammatory immune responses [55] and can also lead to apoptosis [56], both of which would have anti-tumor effects. Future studies will need to examine specific mechanisms of DNA damage by pFUS, which could include generation of reactive oxygen species (ROS) through pFUS-induced cytosolic Ca^2+^ fluxes [57]. We have demonstrated that pFUS generates Ca^2+^ transients by activating mechanically sensitive ion channels in the plasma membrane, which leads to depolarization of voltage-gated calcium channels [58]. While these exposures did not generate cavitation forces, intracellular Ca^2+^ fluxes have also been reported from inertial cavitation mechanical forces [59].

In the current study, the effects of pFUS in different size tumors demonstrates heterogeneity in the TME response. There was little overlap within or between the two tumors with regard to the molecular response on different days 24 h after sonication. Following pFUS, there was significant slowing of B16 and 4T1 tumor growth compared to naïve controls; however, in some cases, the variable expression of pro-inflammatory and anti-inflammatory CCTFs and CAMs does not explain the relationship between these two independent results. The proteomic measures used to interrogate the TME at each time point post-pFUS limited the ability to further identify relevant factor(s) that contributed to the slowing tumor growth. Moreover, the growth and proteomic changes in the TME in the naïve control tumors may also depend on epigenetic factors associated with tumor proliferation, further complicating the interpretation of our results.

This study underscores the possible inherent difficulty in comparing therapeutic outcomes that are initiated in tumors of disparate sizes and possibly location across studies based on proteomic analysis of the TME. When the CCTF and CAM changes post-pFUS focused on the TME, FACS analysis provided the opportunity to monitor the systemic immune response in the Sp, regional LN, and following sonication on days 1, 5, and 9. The multi-directional trafficking of immune cell populations to or from Sp, LNs, and each tumor type was highly variable and did not appear to coincide with the specific CCTFs and CAMs in the TME. The presence of DNA damage in both naïve and treated tumors could have influenced the response of immune cell populations within the host. There were also significant differences in the immune cell populations in the Sp, LNs, and tumors when comparing the naïve control tumors at each time point. In the B16 sonicated tumors, significant increases in TUNEL staining and the localization of tumor-infiltrating CD4^+^ and CD8^+^ T-cells was detected on day 2; however, there were relatively few differences in immune cell populations in the tumors, Sp, and LNs compared to controls (Figure 3A). The most dramatic changes in cell populations occurred on day 10 in the LNs in which there were significant fold changes in T-cells, macrophages, and DC population. There was a greater percentage of T_h_ and T_cyt_ along with the decrease in MDSCs and DCs along with checkpoint receptors and ligands in the tumors. By contrast, 4T1 tumors exhibited inconsistent immune cell profiles at the three locations. There were significant increases in TUNEL positivity and localization levels of CD4^+^ and CD8^+^ cells on day 10, which corresponded to an increased detection of anti-tumor and immunosuppressive cells within the treated 4T1 tumors compared to controls (Figure 3B). The relative lack of changes in the Sp and LN immune cell populations also did not follow a set pattern at the three time points. We have previously reported shifts in the immune cell population in Sp following pFUS in B16 (~8 mm in size) that would correspond approximately to the day 6 tumor. We did not observe an immune cell response and changes in PD1 and CTLA4 within the TME in the current study on day 6 [41]. The shedding of tumor antigen load in the circulation may be responsible for the changes in immune cell types in differently sized tumors. It is unknown whether the timing of sonication in relation to the tumor size was responsible for the observed Sp and tumor immune cell profiles [41]. In this study, the larger sonicated day 9 4T1 tumors would contribute to the increased tropism of the immune cell populations into the tumor, especially with the increased expression of chemoattractants and pro-inflammatory CCTFs in the TME. As FACS analysis for PDL1, PD1, and CTLA4 was performed in the tumor without isolation of the possible cell types, it is unclear whether determining if detection at different time points is a function of changes in the expression of these markers on the tumor or immune cell populations. Nevertheless, the results of the current study emphasize the temporal differences of immune cells trafficking in tumors of various sizes following pFUS sonication.

Cellular and biological immunotherapies have generated great interest over the last years as an adjuvant treatment strategy by eliciting an enhanced anti-tumor immune response within the TME, thereby shifting the balance from cold to hot tumors [60]. However, in view of the general consensus correlating the effectiveness of immunotherapeutic treatments to the presence of a baseline immune response [61], this approach may require the addition of interventions, such as radiation therapy, radiofrequency ablation, or HIFU, to achieve a more robust therapeutic response in solid tumors [27,47,60]. Given the vast number of treatment options, it will be necessary to propose methods to normalize the cancer responsiveness when non-ablative techniques are implemented at different growth phases of the tumors. One approach that can relate histological classification along the spectrum from cold to hot tumors has been the implementation of the immune-score [62,63,64,65,66], standardizing and comparing the distribution (i.e., peripheral, central, or a combination of both) and density of primarily cytotoxic T cells within histological sections within a tumor. An ‘immune contexture’ was proposed later on to predict treatment response by including additional immune variables associated with the nature, density, immune functional orientation, and distribution of immune cells within the tumor [63,66]. In the future, temporal proteomic and cellular responses to non-ablative pFUS to different tumor types may provide additional information regarding the mechanisms and best implementation of the effects of ultrasound to result in anti-tumor outcomes. However, the mechanistic underpinnings of the reported observations remain unknown. Slowed tumor growth following pFUS could be functions of increased apoptosis, reduced cell proliferation, or some combination of both. Future studies will begin to address these potential dynamics.

There are several limitations of this study that need to be addressed. The proteomic and immune cell response following pFUS and samples were obtained 24 h post-treatment for molecular and FACS analysis. As each focal spot within the tumor was treated for a total of 1 s of mechanical acoustic radiation force, future studies should investigate whether the length of sonication exposure burst would influence the molecular and cellular changes in the TME and stimulate a more robust systemic immune response. Monitoring the proteomic responses to pFUS within the first 24 h time point may provide a window into the inflammatory profile within the TME and balance the differences between cold and hot tumors. Treated tumors obtained at later time points for immune cell populations may provide a better understanding of the long-term impact of pFUS on the influence of the balance of the TME. In addition, it is still unclear whether the current results with B16 and 4T1 tumors would translate to other murine flank tumors or metastasis with different growth patterns [67]. Further investigations of the mechanical effects of pFUS on TME may benefit from the isolation of different cell populations and exosomes from variously sized tumors and performing single-cell RNA-seq or other molecular analyses in order to understand how the changes in the TME proceed with disease progression. The mechanotransducive effects of pFUS on CCTF, CAM, and immune cell profiles should also be as assessed in patient-derived tumor xenograft (PDX) models, in which cancer cell expansion would grow in physiologically relevant changes to the TME that mimic human malignancy.

## 4. Materials and Methods

### 4.1. Cell Culture

Melanoma B16 and 4T1 breast cancer cell lines were purchased from American Type Culture Collection (ATCC Manassas, VA, USA). These cell lines were maintained in Dulbecco’s modified Eagle’s medium (Gibco, Life Technologies, Grand Island, NY, USA) supplemented with 10% fetal bovine serum and 1% penicillin streptomycin (Gibco, Life Technologies, Grand Island, NY, USA) and were cultured at 37 °C in a humidified atmosphere at 5% CO_2_. At 75–90% confluency, cells were passaged using TyrpLE Express Enzyme (Gibco, Life Technologies, Grand Island, NY, USA).

### 4.2. Mouse Xenograft B16 and 4T1 Tumor Models

Animal experiments were approved by the Animal Care and Use Committee at our institution (Ethic code: 17736-2011). Animals’ procedures complied with the National Research Council’s Guide for the Care and Use of Laboratory [68]. In addition, 6–8 week old C57BL/6 (total *n* = 84) and BALB/c (total *n* = 84) female mice were purchased from the Jackson Laboratory (Bar Harbor, ME, USA). Depilatory cream was used to removed mice hair on both legs one day before tumors were inoculated or treated with pFUS. Tumors were induced by the subcutaneous injection of 10^6^ B16 or 4T1 cells suspended in 100 μL phosphate buffered saline (PBS) to both flanks of C57BL/6 or BALB/c mice, respectively. Tumor measurements were done externally using a digital caliper, and its volume (V) was calculated by the following formula:V = (ab^2^)/2,(1)
where a = the smallest diameter and b = the perpendicular diameter, and was monitored daily. All mice were anesthetized with 1.5–3% isoflurane in 100% O_2_ during the experimental procedures or tumor measurements. Tumors were permitted to grow to an average diameter of 5 mm before initiating pFUS treatment. This diameter was defined as a starting point (day 1) for natural history growth (Figure 6A) or pFUS treatment (Figure 6B) of xenograft B16 and 4T1 tumors. In natural history experiments, B16 or 4T1 tumors were harvested every other day (*n* = 5/time point) and proteomic analysis was performed (Figure 6A).

### 4.3. pFUS Treatment

Ultrasound-guided pFUS (VIFU 2000 Alpinion Medical Systems, Bothell, WA, USA, www.alpinionusa.com) was administered to tumors in degassed water at 37 °C with a single-element transducer operating at 1.15 MHz and a PNP of 6 MPa (mechanical index = 5.6; I_SATP_ = 2683 W/cm^2^) which had been previously shown to be non-ablative [41]. The entire tumor volume was sonicated with an elemental spacing of 2 mm between points, 10% duty cycle, and a pulse repetition frequency of 5 Hz, US burst 20 ms. Six-8 raster points in a 2 × 3 pattern with an elemental spacing of 2 mm were used to treat tumors. pFUS treatments were initiated on days 1, 5, or 9 after the tumor size reached ~5 mm in diameter (Figure 6B). Non-sonicated separate tumors were used as time-matched controls. Tumor samples were collected 24 h after pFUS treatment on days 2, 6, and 10 for proteomic analysis, FACS analysis, and histology including analysis for DNA damage. FACS analysis was also used to determine immune cell populations in Sp and regional LNs.

### 4.4. Proteomic Analyses

Tumors were harvested and instantly flash-frozen through immersion in liquid nitrogen and stored at −80 °C until use. Tumor samples were homogenized in cell lysis buffer containing Tris-buffered saline (pH 7.5) supplemented with protease inhibitor cocktail (Roche, Mannheim, Germany) and 0.1% Tween 80 using 1.0 mm zirconia beads (Biospect, Bartlesville, OK, USA) by Mini-bead Beater (Biospect, Bartlesville, OK, USA). Samples were then centrifuged at 14,000 rpm for 20 min at 4 °C, and the supernatant of each sample was collected. A protein concentration of 1 mg/ml for each sample was determined by bicinchoninic acid assay (PierceTM, Thermo Scientific, Rockford, IL, USA). Proteomic analysis for pFUS-treated and untreated tumors was measured using a Mouse Cytokine Chemokine Magnetic Bead Panel (MCYTMAG-70-PX32, MILLIPLEX^®^ MAP Kit, Merk Millipore, Lexington, MA, USA) with the Bio-Plex 200 System (Bio-Rad, Hercules, CA, USA), according to the manufacturer’s protocol. In addition, enzyme-linked immunosorbent assays (ELISA) for intracellular adhesion molecule-1 (Mouse ICAM-1/CD54; DY796), vascular cell adhesion molecule (Mouse VCAM-1/CD106; DY643), TGF-β1 (MB100B/SMB100B), and IFNγ (DuoSet; DY485; R&D Systems, Minneapolis, MN, USA) were conducted using a spectrophotometric plate reader (Spectra Max M5, Molecular Devices, Sunnyvale, CA, USA). Fold changes in protein expressions were calculated as a ratio between mean concentrations of the detected CCTFs and CAMs compared to the average concentration detected from untreated tumors on day 1 (Figure 6A) or to separated control group time-matched untreated tumors (Figure 6B).

### 4.5. Flow Cytometric Analyses

Twenty-four hours post-sonication on days 1, 5, and 9, the Sp, inguinal LNs, and tumors (*n* = 6 mice/tumor type/time point) were harvested (days 2, 6, and 10) and dissociated into single-cell suspensions. Non-sonicated tumors served as time-matched controls. Single-cell suspensions of B16 or 4T1 tumors were obtained by mechanical disruption of the tissue followed by enzymatic digestion with HBSS (Gibco, Life Technologies, Grand Island, NY, USA) supplemented with 2% fetal bovine serum, 1% penicillin streptomycin, and 1% collagenase (Type IV; Sigma-Aldrich, St. Louis, MO, USA) for 60 min at 37 °C. Following digestion, tumors were mechanically agitated, forced through 70 µm cell strainers (BD Biosciences, San Jose, CA, USA), and fixed for 15 min using 4% paraformaldehyde. Cell viability was not quantified using FACS, due to the numbers of samples processed for multiple markers on any day, requiring cell fixation and analysis over subsequent days. Spleens and tumor-draining inguinal LNs suspended in ammonium-chloride-potassium (ACK) lysing buffer (Invitrogen, Carlsbad, CA, USA) were minced, mashed, and passed through 70 μm cell strainers. Cells were then washed in PBS, fixed using 4% paraformaldehyde, and stored in PBS at 4 °C. For immunostaining, 1 million cells from Sp, LN, or tumor samples were initially treated with Fc-receptor-blocking antibody (purified anti-mouse CD16/32 antibody, BioLegend, San Diego, CA, USA) in staining buffer (0.5% BSA, 2 mM EDTA, 1X PBS) for 10 min on ice to reduce non-specific immunofluorescent staining. Cells were then stained with the following fluorescently labelled antibodies (BioLegend, San Diego, CA, USA) diluted in staining buffer for 45 min on ice: FITC anti-mouse CD3 (clone 17A2), FITC anti-mouse F4/80 (clone BM8), FITC anti-mouse CD45 (clone 30-F11), PE anti-mouse CD25 (clone PC61), PE anti-mouse CD8a (clone 53-6.7), PE anti-mouse CD206 (clone C068C2), Alexa Fluor 488 anti-mouse/human CD45R/B220 (clone RA3-6B2), PE anti-mouse CD11c (clone N418), PE anti-mouse CD11b (clone M1/70), APC anti-mouse CD4 (clone RM4-5), APC anti-mouse CD86 (clone GL-1), APC anti-mouse Ly-6G/Ly-6C (Gr-1) (clone RB6-8C5), APC anti-mouse CD152 (clone UC10-4B9), Alexa Fluor^®^ 647 anti-mouse CD335 (NKp46) (clone 29A1.4), PE anti-mouse CD274 (clone 10F-9G2), and Alexa Fluor^®^ 647 anti-mouse CD279 (PD1) (clone 29F.1A12). Matched fluorescently labelled isotype controls were used (Appendix A). Next, the stained cells were washed with PBS, resuspended in 200 µL PBS, and loaded onto V-shaped 96-well plates. FACS data were collected on a BD Accuri^TM^ C6 Cytometer equipped with BD Accuri^TM^ C-Sampler (BD Biosciences, San Jose, CA, USA) and was processed using FlowJo (FlowJo LLC, OR, USA) software. Then, 10,000 events were acquired from each sample 

### 4.6. Histology and Immunohistochemistry 

Mice euthanized for histological evaluation and harvested tumors were submerged in 4% paraformaldehyde in PBS (pH 7.4) for 24 h. B16 and 4T1 tumor tissue were then embedded in paraffin blocks and sectioned serially at a 6 µm thickness. Following 1 h incubation at 65 °C, sections were deparaffined with xylene and accompanied by sequential ethanol hydration. Tumor sections (*n* = 3 sections/tumor type/time point) were stained with hematoxylin and eosin (H and E) or fluorescein-based staining. For H and E staining, slides were treated with hematoxylin solution for 4 min and then washed with tap water for 1 min. The samples were treated with bluing reagent for 4 min, then immediately washed in tap water. Samples were rinsed in 95% ethanol (2 × 2 min) and counterstained in eosin solution for 3 min. The samples were dehydrated through graded ethanol solutions (95% and 100% 2 × 2 min), cleared in xylene (2 × 2 min), and were then mounted with Cytoseal 60 (VWR International, Philadelphia, PA, USA). Fluorescence staining was performed following antigen retrieval in citric acid-based (pH = 6.0) buffer (Vector Laboratories, Burlingame, CA, USA) using a pressure cooker at 100 °C for 30 min and incubating the samples with 20% horse serum and 0.2% Triton X-100 in PBS for 1 h. Slides were incubated overnight with the primary antibodies rabbit anti-mouse CD4 (clone EPR19514, Abcam, Cambridge, UK) or rat anti-mouse CD8 (clone 4SM15, Fisher Scientific, Pittsburgh, PA, USA) in a humidified chamber at room temperature. Next, biotinylated goat anti-rabbit (1:200) was followed by Cy3-conjugated streptavidin (Jackson Immunoresearch Laboratories, West Grove, PA, USA) incubation, both at room temperature for 1 h. Cell nuclei were fluorescently stained with DAPI (Molecular Probes Inc., Eugene, OR, USA), and mounted sections were visualized using a ScanScope CS (20× air objective, Aperio Technologies, Vista, CA, USA) under identical settings. For quantitative analysis of CD4^+^ and CD8^+^ localization, the numbers of fluorescence-positive cells within the center or margin areas of tumors were counted under blinded conditions and compared to the fluorescence-positive cells count within their corresponding control untreated tumors.

### 4.7. DNA Damage Assay

Histological paraffin sections were also used to investigate relative DNA damage in the cell death response to 6 MPa pFUS treatment (Roche Applied Science, Indianapolis, IN) by the fluorescein-based in situ cell death terminal UTP end-nick ligase (TUNEL) detection kit (Roche Applied Science, Indianapolis, IN, USA). Paraffin sections were deparaffinized, sequentially hydrated, and pretreated with heat-induced antigen retrieval, as described earlier. Sections were permeabilized using 0.1% Tritonx-100 in PBS and incubated with TUNEL reaction mixture for 1 h at 37 °C. Samples were mounted following cell nuclei staining and visualized using identical exposure times across tissues. To assess patterns of DNA damage, 3 sections of each tumor type from *n* = 3 mice per group were analyzed under blinded conditions using ImageJ software (Wayne Rasband, NIH, Bethesda, MD, USA). Fluorescence intensity levels above a set of thresholds were normalized by the total number of fluorescent-positive nuclei found within the same field of view.

### 4.8. Statistical Analysis 

Data are presented as mean ± standard deviation and the number of animals of each experiment indicated accordingly. All experiments were analyzed using Prism 8, GraphPad Software, Inc. (La Jolla, CA, USA). A *p* value < 0.05 was considered statistically significant. One-way analysis of variance (ANOVA) with Bonferroni post-hoc tests was used for multiple comparison, and a t-test with the Mann–Whitney test was used for unpaired non-parametric comparison.

## 5. Conclusions

In summary, we demonstrate the importance of evaluating the changes in CCTF, CAM, and immune cell population caused by pFUS mechanical forces in two different types of tumors. Our results emphasize the profound magnitude of tumoral heterogeneity not only between tumor types but also through progressive stages of similar primary tumors. Non-ablative pFUS-induced effects on the TME and immune cell populations provide additional novel insights into the mechanisms associated in shifting the balance between cold and hot tumors and allow for potentially being harnessed for developing other approaches in cancer immunotherapy.

## Figures and Tables

**Figure 1 cancers-12-00350-f001:**
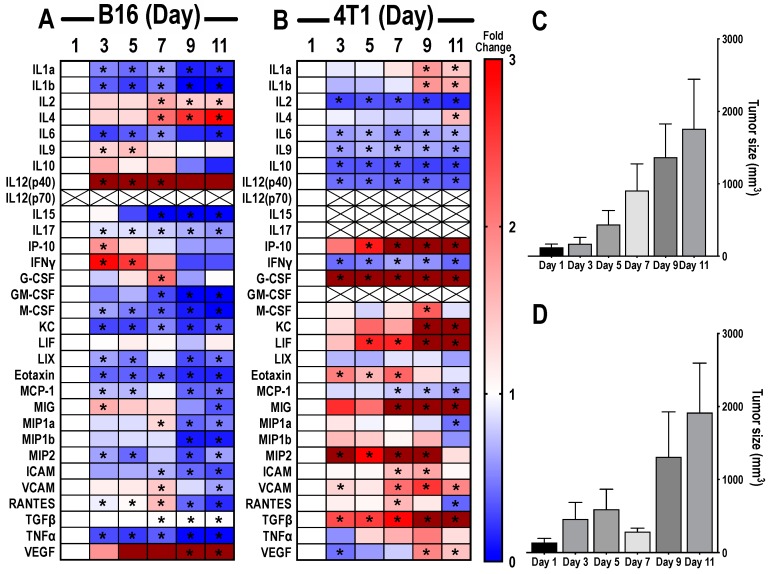
Proteomic changes of cytokines, chemokines, and trophic factors (CCTFs) over time in tumor microenvironment (TME) of mouse xenograft B16 melanoma (**A**) or 4T1 breast cancer (**B**). Heat maps depict the calculated ratio between the mean concentration of the detected CCTFs in each time point to the average concentration detected on day 1. Blue represents fold changes less than 1. Red represents 1–3 fold changes. Dark red represents fold changes >3.1. Asterisks indicate statistical significance (*p* < 0.05) identified by one-way analysis of variance (ANOVA) test; tumor size volumes of mouse xenograft B16 melanoma (**C**) or 4T1 breast cancer (**D**) models were determined 3, 5, 7, 9, and 11 days after reaching ~5 mm size in diameter (day 1 tumors).

**Figure 2 cancers-12-00350-f002:**
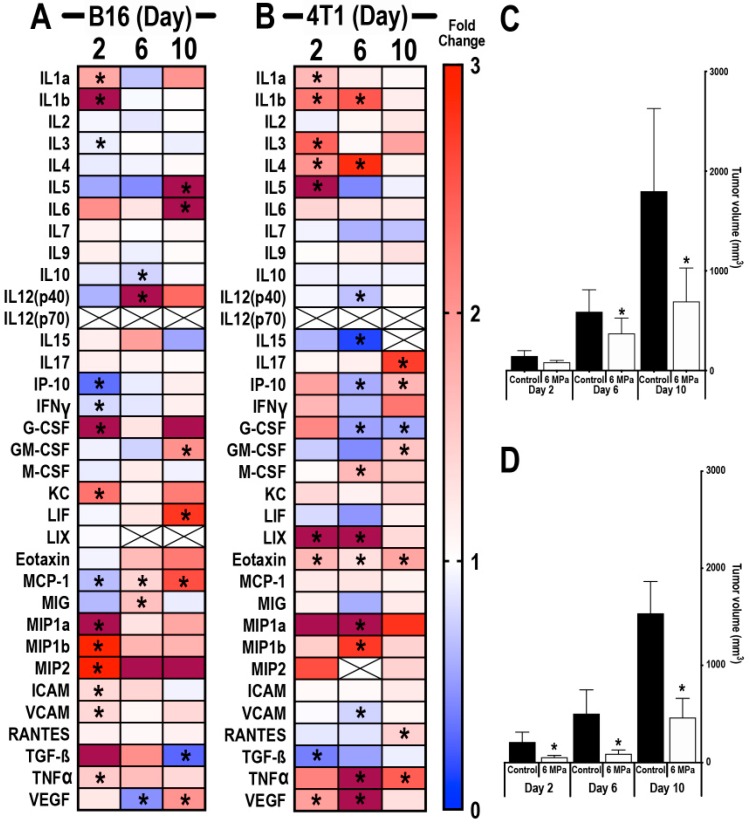
Proteomic response of B16 or 4T1 tumors following pulsed focused ultrasound (pFUS) treatment on different growth progressions. Heat map depicting fold changes in CCTFs 24 h post-sonication of mouse B16 (**A**) or 4T1 (**B**) tumors on days 2, 6, or 10. Tumors from non-sonicated control mice on days 2, 6, and 10 served as time-matched controls. Protein levels are presented as fold changes in relation to time-matched non-sonicated control values. Blue represents fold changes less than 1. Red represents 1–3 fold changes. Dark red represents fold changes >3.1. Tumor size volumes of mouse xenograft B16 melanoma (**C**) or 4T1 breast cancer (**D**) model; asterisks indicate statistical significance (*p* < 0.05) identified by ANOVA test.

**Figure 3 cancers-12-00350-f003:**
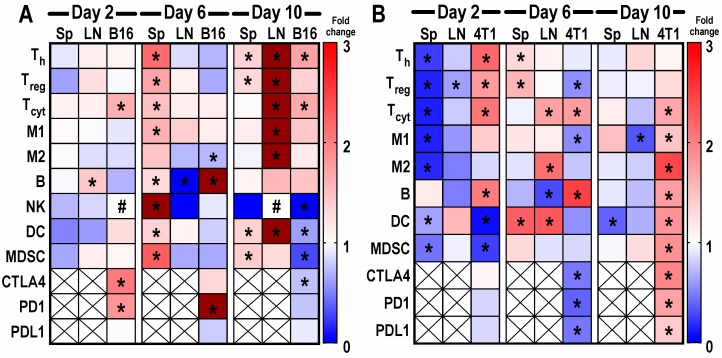
Heat maps depicting fold changes of immune cell profiles and checkpoint receptor/ligands from mice with B16 melanoma (**A**) or 4T1 breast (**B**) tumor, spleen (Sp), and lymph node (LN) that were harvested 24 h following pFUS treatment on days 1, 5, or 9. Day 1 tumors were ~5 mm in size. Non-sonicated tumors harvested on day 2, 6, or 10 served as time-matched controls. Flow cytometry analysis was performed by staining fresh fixed immune cells (CD8^+^ T_cyt_, CD4^+^ T_h_, T_reg_, natural killer (NK) cells, dendritic cells (DCs), F4/80 macrophages (M1 and M2), myeloid suppressive cells along with cytotoxic T-lymphocyte associated protein 4 (CTLA4), programmed cell death protein 1 (PD1), and programmed death ligand 1 (PDL1)) in the tumors. Blue represents fold changes less than 1. Red represents 1–3 fold changes. Dark red represents fold changes higher than 3. Asterisks indicate statistical significance (*p* < 0.05) identified by ANOVA test. Pound indicates undetected values.

**Figure 4 cancers-12-00350-f004:**
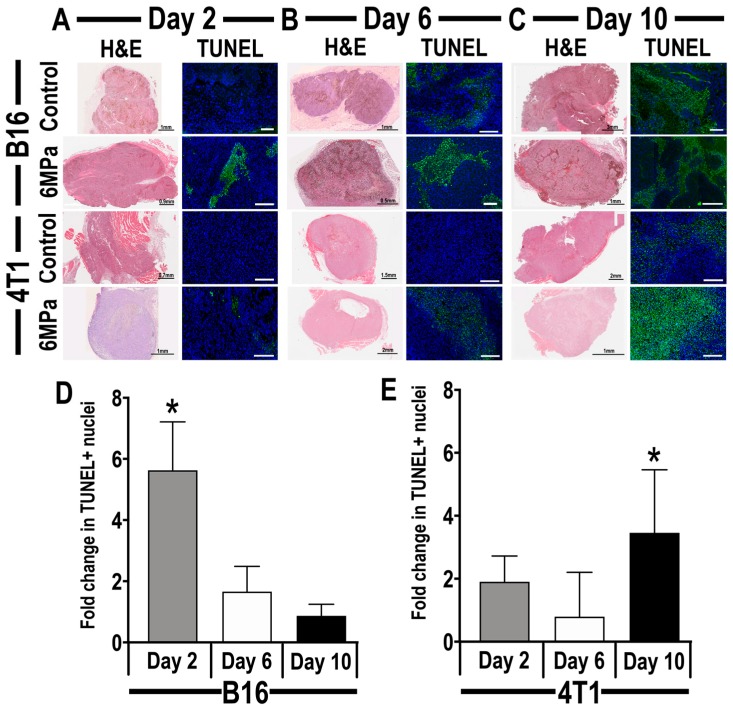
Representative histological sections from mouse xenograft B16 melanoma tumor (**upper panel**) or 4T1 breast cancer (**lower panel**), 24 h following treatment with 6 MPa pFUS on days 2 (**A**), 6 (**B**), or 10 (**C**). (**A**–**C**) **left**: Representative hematoxylin and eosin (H and E)-stained; (**A**–**C**) **right**: DNA-damaged cells (green) in mouse xenograft tumor models (scale 200 µm); quantitative analysis of time-related changes in the number of DNA-damaged cells per field of view on B16 (**D**) or 4T1 (**E**) tumors, 24 h following pFUS treatment. Asterisks indicate statistical significance (*p* < 0.05) identified by ANOVA test.

**Figure 5 cancers-12-00350-f005:**
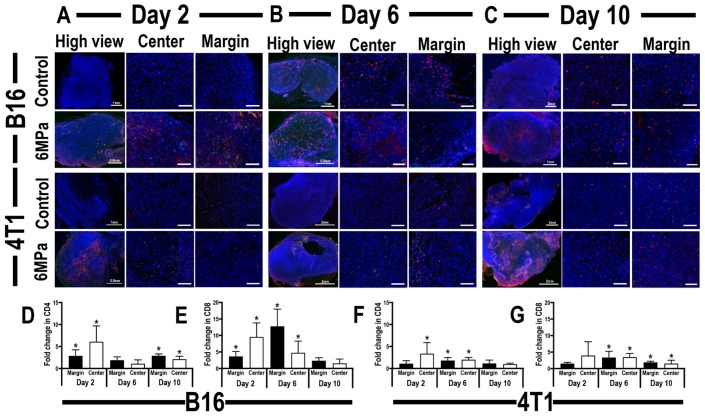
Representative immunohistochemistry staining of CD4 (green), CD8 (red), and DAPI (blue) in tumor sections of mouse xenograft B16 melanoma tumor model (**upper panel**) or 4T1 breast cancer model (**lower panel**), after reaching 5 mm in diameter, 24 h post-pFUS (**second and fourth rows**) or untreated tumors (**first and third rows**). High-view image of B16 or 4T1 tumors on day 2 (**A**, **left**), day 6 (**B**, **left**), or day 10 (**C**, **left**); scale 100 µm. Lower view of center tumor areas in B16 or 4T1 tumors on day 2 (**A**, **middle**), day 6 (**B**, **middle**), or day 10 (**C**, **middle**); scale 100 µm. Lower view of tumor periphery areas in B16 or 4T1 tumors on day 2 (**A**, **right**), day 6 (**B**, **right**), or day 10 (**C**, **right**); scale 100 µm. Quantitative analysis of time-related changes in CD4^+^ (**D**) or CD8^+^ (**E**) cell localization within center or margin areas of B16 tumors. Quantitative analysis of time-related changes in positive CD4^+^ (**F**) or CD8^+^ (**G**) cell localization within center or margin areas of 4T1 tumors. Asterisks indicate statistically significant elevations (*p* < 0.05; ANOVA) between the pFUS-treated group to a time-matched untreated control.

**Figure 6 cancers-12-00350-f006:**
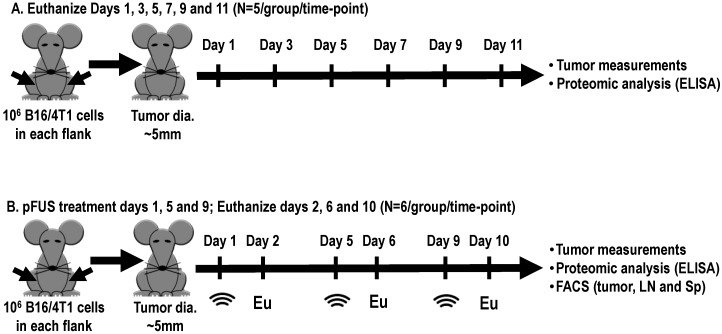
Schematic diagram of the experimental setups. The starting point for all experiments was defined once tumor sizes reached ~5 mm in diameter. (**A**) Natural history growth of xenograft B16 or 4T1 tumors, harvested every other day without treatment for molecular analysis. (**B**) Tumors were treated with 6 MPa pFUS on days 1, 5, or 9, and tumors, spleens (Sp), or regional lymph nodes (LNs) were harvested 24 h for molecular, flow cytometry (FACS), and histology; pFUS: Pulse focus ultrasound; Eu: Euthanize.

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
