# Peer review of "The Impact of Focused Ultrasound in Two Tumor Models: Temporal Alterations in the Natural History on Tumor Microenvironment and Immune Cell Response"

_cancers, 2020, doi:10.3390/cancers12020350_

Round 1
Reviewer 1 Report
In this study, the authors investigated the effect of pFUS (i.e. a non-ablative focused ultrasound modality) on the tumor microenvironment (i.e. cytokines and immune cells). Two cancer cell lines were used: B16 (Melanoma) and 4T1 (Breast) to create xenograft models in C57BL/6 and BALB/c. In general, there was a shift towards and immunosuppressive/Cold phenotype within the tumor microenvironment. However, following pFUS, there was a decrease in tumor growth rate and B16 tumors showed dynamic innate and adaptive immune modulation at early time points in contrast to 4T1 tumors which showed this at later time points.
Results
Figure 1
Figure A/B: How do the authors explain the opposing trends (i.e., increased then decreased, or decreased then increased) in certain molecules? For example, IFN-γ in B16 tumors and VEGF in 4T1 tumors? What might explain this differential response? Figure D: Authors describe the growth patterns of both B16 and 4T1 cancers as exponential, but growth pattern in 1D appears to be bimodal instead?
Figure 3
Figure 3A: What would explain the overall trends detailed by the authors between immune cell profiles in the spleen, LN, and tumor? For example, what accounts for the splenic response (influx of cells) in day 6 (T cells, macrophages), and a lymph node response in Day 10?
Figure 4
Given that in B16 tumors, only day 2 exhibited greater apoptosis, and in 4T1 tumors, only day 10 exhibited greater apoptosis, how would the authors describe the decrease in tumor size (due to pFUS) indicated previously in Figure 2? Is this due to increased apoptosis, decreased proliferation, or decreased viability? The authors may want to consider whether the decreased tumor burden may also arise due to reduced cell viability/proliferation
Page 3
2.1.1. “Proteomic Profiling Dduring the Natural History Growth of Naïve Tumors”
Please correct the typo “during”
“There were significant decreases (p<0.05 ANOVA) in expression of IL2, IL6, IL9, IL10, IL12p40, TGFβ and IFNγ, and increased expression for G-CSF, VCAM and TGFβ over all days compared to tumors on day 1.”
TGFβ is mentioned as both increased and decreased where as heat map shows significant increase 1- 3 fold and >3.1
Page 5
“Examination of CCTF and CAM in naïve controls at days 2, 6, and 10 revealed similar appearance of the heat map when compared to the natural history study which would support that there is overlap in the changes in TME with tumor size (data not shown).”
Can this data be provided, maybe in the suplementary files.
Page 6
“TME by day 10, which may provide explanation to the decrease in pFUS treated tumor volume (Figure 3D).” It should be (Figure 3B)
Discussion:
Is there a way the results can also be interpreted in the context of an immune contexture or in relation to an immune score as discussed by the authors in the conclusion Could a schematic be added to give an overview of the proposed mechanisms of action based on the data Can the authors please add a proposed mechanism of action of how pFUS is working based on conceptual theories with supported references?
General Comments:
Can the authors present Figure 4 and 5 so they are the same layout/format - Control and 6MPa are on the top? Maybe consider also expressing all the results are a relative fold increase or show the control values for both the TUNEL and CD4/CD8
Since authors used 1-way ANOVA with Bonferroni post-hoc, how do they account for the larger rate of false positives that occur with multiple testing (since alpha is still < 0.05)? For example, the first few figures include >30 pairwise tests.
The authors suggest that the anti-tumor response may lead to decreased overall tumor burden. Although the size-reducing effects of pFUS have been illustrated by the present study, did the authors evaluated whether pFUS affects the degree of tumor invasion and metastasis?
Author Response
Comments and Suggestions for Authors
In this study, the authors investigated the effect of pFUS (i.e. a non-ablative focused ultrasound modality) on the tumor microenvironment (i.e. cytokines and immune cells). Two cancer cell lines were used: B16 (Melanoma) and 4T1 (Breast) to create xenograft models in C57BL/6 and BALB/c. In general, there was a shift towards and immunosuppressive/Cold phenotype within the tumor microenvironment. However, following pFUS, there was a decrease in tumor growth rate and B16 tumors showed dynamic innate and adaptive immune modulation at early time points in contrast to 4T1 tumors which showed this at later time points.
Results
Figure 1
Figure A/B: How do the authors explain the opposing trends (i.e., increased then decreased, or decreased then increased) in certain molecules? For example, IFN-γ in B16 tumors and VEGF in 4T1 tumors? What might explain this differential response? Figure D: Authors describe the growth patterns of both B16 and 4T1 cancers as exponential, but growth pattern in 1D appears to be bimodal instead?
Response: Opposing trends represent changes in CCTF/CAM expression over time as a function of the natural history of each tumor model, highlighting a major point of the manuscript that tumor microenvironments differ over time and therefore could affect treatment effects/outcomes at different times. In the growth pattern for 4T1, the day 7 measurement was not statistically significantly lower than the previous measurement on Day 5 and is likely a reflection of the number of animals measured in this particular experiment. Should this experiment have been repeated multiple times, we would not expect to measure a statistically significant bi-phasic pattern. Nevertheless, the descriptions of growth patterns were rephrased.
Figure 3
Figure 3A: What would explain the overall trends detailed by the authors between immune cell profiles in the spleen, LN, and tumor? For example, what accounts for the splenic response (influx of cells) in day 6 (T cells, macrophages), and a lymph node response in Day 10?
Response: Different trends in response to pFUS at different time points likely stem from the natural histories of the tumor types (see response above). Given that the disease evolves, from a molecular and cellular standpoint, over the course of the experiment, Figure 3 demonstrates that pFUS responses will differ depending upon when in the natural history course pFUS is applied.
Figure 4
Given that in B16 tumors, only day 2 exhibited greater apoptosis, and in 4T1 tumors, only day 10 exhibited greater apoptosis, how would the authors describe the decrease in tumor size (due to pFUS) indicated previously in Figure 2? Is this due to increased apoptosis, decreased proliferation, or decreased viability? The authors may want to consider whether the decreased tumor burden may also arise due to reduced cell viability/proliferation
Response: TUNEL only measures DNA damage and is not a reflection of cellular apoptosis per se. DNA damage can be a potent damage-associated molecular pattern (DAMP) that stimulates immune responses that can alter both apoptosis and proliferation rates. Our data do not specifically implicate if reduced tumor growth results from one process dominating over the other or if some combination of both is responsible (pages 10-11; lines 410-413). These questions will be addressed in future studies.
Page 3
2.1.1. “Proteomic Profiling Dduring the Natural History Growth of Naïve Tumors” Please correct the typo “during”
Response: This has been corrected.
“There were significant decreases (p<0.05 ANOVA) in expression of IL2, IL6, IL9, IL10, IL12p40, TGFβ and IFNγ, and increased expression for G-CSF, VCAM and TGFβ over all days compared to tumors on day 1.”
TGFβ is mentioned as both increased and decreased whereas heat map shows significant increase 1- 3 fold and >3.1
Response: We thank the reviewer for this comment, the manuscript text and supplemental figure 1 were rectified and revised.
Page 5
“Examination of CCTF and CAM in naïve controls at days 2, 6, and 10 revealed similar appearance of the heat map when compared to the natural history study which would support that there is overlap in the changes in TME with tumor size (data not shown).” Can this data be provided, maybe in the supplementary files.
Response: We would like to thank the reviewer for their suggestion. We have included heat maps for B16 (Supplemental Figure S5A) and 4T1 (Supplemental Figure S6A) illustrating the ratio between the mean concentration of the detected CCTF and CAM values in naïve controls at days 6, and 10 to the average concentration detected on day 2.
Page 6
“TME by day 10, which may provide explanation to the decrease in pFUS treated tumor volume (Figure 3D).” It should be (Figure 3B)
Response: This typo has been corrected.
Discussion:
Is there a way the results can also be interpreted in the context of an immune contexture or in relation to an immune score as discussed by the authors in the conclusion Could a schematic be added to give an overview of the proposed mechanisms of action based on the data Can the authors please add a proposed mechanism of action of how pFUS is working based on conceptual theories with supported references?
Response: The references to immune-score apply only to T cell phenotyping and distribution, but one does not currently exist for CCTF and CAM in the microenvironment. We are, however, exploring the development of one for future studies. While this study characterizes microenvironmental and immune changes due to pFUS in the context of the natural histories of each tumor, we do not specifically understand the biological mechanism beyond upregulated pro-inflammatory CCTF and CAM resulting in altered immune cell trafficking. We have noted this in the discussion (page 9; lines: 341-349) and that future studies will seek to uncover more detailed cellular and molecular mechanisms behind pFUS effects.
General Comments:
Can the authors present Figure 4 and 5 so they are the same layout/format - Control and 6MPa are on the top? Maybe consider also expressing all the results are a relative fold increase or show the control values for both the TUNEL and CD4/CD8.
Response: These figures have been altered for consistency.
Since authors used 1-way ANOVA with Bonferroni post-hoc, how do they account for the larger rate of false positives that occur with multiple testing (since alpha is still < 0.05)? For example, the first few figures include >30 pairwise tests.
Response: Separate ANOVAs were performed for each CCTF/CAM. Thus, each CCTF/CAM only undergoes 5 comparisons in Figure 1 and 3 comparisons in Figure 2. We feel that the Bonferroni correction for multiple tests is sufficient with this number of tests.
The authors suggest that the anti-tumor response may lead to decreased overall tumor burden. Although the size-reducing effects of pFUS have been illustrated by the present study, did the authors evaluated whether pFUS affects the degree of tumor invasion and metastasis?
Response: The experimental time courses (no later than 14 days following tumor inoculation) presented here are insufficient to thoroughly examine invasion and metastasis. However, these are critical questions to be answered in this field and we plan to conduct future studies to do so.
Reviewer 2 Report
The author investigated the immunomodulation effects in CCTF, CAM and immune cell population caused by pFUS mechanical forces in two different types of tumors. The topic of this study is challenging and interesting.
Discuss DNA double-strand breaks induced by pFUS according to inertial cavitation activity. How much power did you set with a spatial average, temporal peak intensity (ISATP) in pFUS? How many minutes did the pFUS exposure last? The authors need to show the illustration of intratumoral mechanism induced by pFUS.
Author Response
Comments and Suggestions for Authors
The author investigated the immunomodulation effects in CCTF, CAM and immune cell population caused by pFUS mechanical forces in two different types of tumors. The topic of this study is challenging and interesting.
Discuss DNA double-strand breaks induced by pFUS according to inertial cavitation activity. How much power did you set with a spatial average, temporal peak intensity (ISATP) in pFUS? How many minutes did the pFUS exposure last? The authors need to show the illustration of intratumoral mechanism induced by pFUS.
Response: We have included text and references in the discussion to propose potential mechanisms resulting from cytosolic calcium fluxes, which have been observed in cells during pFUS with and without inertial cavitation. We have also reported the ISATP as 2683 W/cm2 (Page 12; lines: 458-459). Per Reviewer 1’s comments, we are unsure about precise mechanisms and will investigate them in future studies.
Reviewer 3 Report
In this study, the authors evaluated the natural history of the chemokines and trophic factors (CCTF), and cell adhesion molecules (CAM) changes in 4T1 and B16 flank tumors over 11 days to determine how the TME was altered with tumor growth. Authors also investigated whether pFUS exposure at different times (days 1, 5, and 9) during tumor growth resulted in molecular, immune cell, and histological changes within the TME.
The manuscript is well written and nicely prepared. The experiments results and analyses are convincing.
I recommend the publication of this manuscript following minor correction.
-Figure 4 & 5: Graphs are small, text is barely visible in the print version.
Author Response
Comments and Suggestions for Authors
In this study, the authors evaluated the natural history of the chemokines and trophic factors (CCTF), and cell adhesion molecules (CAM) changes in 4T1 and B16 flank tumors over 11 days to determine how the TME was altered with tumor growth. Authors also investigated whether pFUS exposure at different times (days 1, 5, and 9) during tumor growth resulted in molecular, immune cell, and histological changes within the TME.
The manuscript is well written and nicely prepared. The experiments results and analyses are convincing.
I recommend the publication of this manuscript following minor correction.
-Figure 4 & 5: Graphs are small, text is barely visible in the print version.
Response: These figures have been altered for clarity.